

# A collaborative inference strategy for medical image diagnosis in mobile edge computing environment

Shiqian Zhang[1,2], Yong Cui[3], Dandan Xu[1,2] and Yusong Lin[2,4]

[1] School of Computer and Artificial Intelligence, Zhengzhou University, Zhengzhou, Henan, China
[2] Collaborative Innovation Center for Internet Healthcare, Zhengzhou University, Zhengzhou, Henan, China
[3] School of Computer and Communication Engineering, Zhengzhou University of Light Industry, Zhengzhou, Henan, China
[4] School of Cyber Science and Engineering, Zhengzhou University, Zhengzhou, Henan, China

Corresponding author
Yusong Lin, yslin@ha.edu.cn

## ABSTRACT

The popularity and convenience of mobile medical image analysis and diagnosis in mobile edge computing (MEC) environments have greatly improved the efficiency and quality of healthcare services, necessitating the use of deep neural networks (DNNs) for image analysis. However, DNNs face performance and energy constraints when operating on the mobile side, and are limited by communication costs and privacy issues when operating on the edge side, and previous edge-end collaborative approaches have shown unstable performance and low search efficiency when exploring classification strategies. To address these issues, we propose a DNN edge-optimized collaborative inference strategy (MOCI) for medical image diagnosis, which optimizes data transfer and computation allocation by combining compression techniques and multi-agent reinforcement learning (MARL) methods. The MOCI strategy first uses coding and quantization-based compression methods to reduce the redundancy of image data during transmission at the edge, and then dynamically segments the DNN model through MARL and executes it collaboratively between the edge and the mobile device. To improve policy stability and adaptability, MOCI introduces the optimal transmission distance (Wasserstein) to optimize the policy update process, and uses the long short-term memory (LSTM) network to improve the model's adaptability to dynamic task complexity. The experimental results show that the MOCI strategy can effectively solve the collaborative inference task of medical image diagnosis and significantly reduce the latency and energy consumption with less than a 2% loss in classification accuracy, with a maximum reduction of 38.5% in processing latency and 71% in energy consumption compared to other inference strategies. In real-world MEC scenarios, MOCI has a wide range of potential applications that can effectively promote the development and application of intelligent healthcare.

## INTRODUCTION

With the rapid development of deep learning technology, deep neural networks (DNNs) have demonstrated significant advantages in medical image analysis. DNNs are capable of automatically learning complex feature representations from a large amount of medical image data and have shown the potential to outperform traditional methods, especially in tasks such as cancer screening, tumor detection and organ segmentation.

Despite the remarkable results achieved by DNN in medical image analysis, the application of DNN in medical image analysis faces major challenges due to the explosion of image data and the increasing demand for accurate diagnosis (*Liu et al., 2024*), whose computational complexity and high-dimensional data processing requirements make it difficult to run a complete DNN model on a single device. In particular, when processing high-resolution images and large datasets, traditional medical devices typically lack sufficient computational resources to support real-time inference. In addition, medical image analysis requires extremely high accuracy and real-time performance, while DNN models typically require a large amount of computational resources, resulting in slower inference and higher latency, making it difficult to meet the demands of real-time medical diagnosis.

Mobile edge computing (MEC) has emerged as a new computing model to solve the problem of insufficient computing resources (*Lin et al., 2023*). By shifting computing power to edge devices closer to users (*Shi et al., 2016*), MEC can effectively reduce data transmission delays and improve real-time performance (*Jedari et al., 2021*). However, MEC solutions also face several challenges:

i. Device performance limitations: Although edge devices have some computational capability, it is still difficult to fully perform the inference task of complex DNN models due to their limited resources.

ii. Communication latency and bandwidth problem: Edge-side collaborative inference often involves communication between multiple devices, especially when the neural network performs model slicing (*Kang et al., 2017*), and transmitting the intermediate layer feature data increases the bandwidth load and latency, affecting the overall performance of the system.

Current collaborative inference methods usually define tasks only by the size of input data and the number of CPU cycles required, ignoring the indivisibility of individual DNN layers, resulting in task assignment that does not fully consider the details and complexity of the inference process. In addition, the multi-agent collaborative inference problem contains a hybrid action space, such as discrete offloading channels and the continuous transmit power, while the existing methods only focus on either a discrete or continuous action space, which makes the existing strategies less adaptive and robust in the face of dynamic network changing conditions and multi-device interference. Especially in the field of medical imaging, the shortcomings of traditional methods become more and more significant, and it is difficult to meet the requirements of efficient inference and high accuracy in practical applications.

To solve the above problems, we propose a DNN edge-end optimized collaborative inference strategy (MOCI) for medical image diagnosis with respect to the specific needs of the medical image diagnosis classification task. The MOCI strategy makes extensive use of edge computing, model segmentation, feature compression, and task offloading and scheduling to achieve efficient distributed inference. Specifically, the pre-training model for medical image classification is divided into two parts, which are executed on the user equipment (UE) and edge servers respectively, and the intermediate feature compression technique is used to reduce the amount of transmitted data and computation delay, and through reasonable scheduling in the edge environment, it ensures that the medical image cloud platform can run efficiently and stably even during peak hours to provide timely medical image analysis services (*Liu et al., 2023*). The main contributions of this article are summarized as follows:

1) We propose a compression method for edge scenarios that uses regularization to prevent overfitting and combines an autoencoder and a quantization module to compress intermediate features, achieving a reduction in latency and energy consumption with less than a 2% loss in classification accuracy.

2) We propose a method based on multi-agent reinforcement learning (MARL), which introduces the optimal transmission distance (Wasserstein) under the original strategy to further constrain the old and new strategies. Meanwhile, in order to effectively cope with the dynamic changes of the network and the complexity of the task, a long short-term memory (LSTM) structure is combined in the criterion network, which takes advantage of the LSTM's ability to process sequential data and improves the accuracy and stability of the criterion network in evaluating state-action pairs.

3) The proposed MOCI strategy can handle multiple agents and use a hybrid action space while accounting for interference between multiple user devices.

The rest of this article is organized as follows. "Related work" presents the related work. "System overview and problem modeling" illustrates an overview of the system and the modeling of the problem, including considerations of the task model, communication model, computational model, and problem description. "System design" is used for system design, including compression features and DNN segmentation programming. "Experiments" design the simulation experiment and present the numerical results. Finally, a summary of our work and future plan is presented in "Conclusion".

## RELATED WORK

Relevant studies on mobile medical image inference can be classified into three categories based on the deployment method: mobile-based inference, edge/cloud-based inference, and collaborative inference.

### Mobile inference

*Zulkifley et al. (2021)* proposed a residual mixing network with a spatial pyramid pooling module to implement an automated pneumonia screening system based on X-ray imaging, and the lightweight network model and residual mixing unit make it suitable for mobile

applications. *Vaze, Xie & Namburete (2020)* used separable convolution for fast medical image segmentation and reduced the number of parameters to facilitate mobile deployment. However, the drawback is that all the above studies are difficult to meet the application scenarios with high real-time requirements. *Gan et al. (2021)* simplified model parameters by model pruning method to reduce model size and computation time, and applied it to carotid artery image segmentation for mobile devices. However, the inference model speed and accuracy still need to be improved. *Garifulla et al. (2021)* implemented a mobile device-based classification of benign and malignant breast cancer using a quantization technique to simplify the network model, but its performance was limited by being limited to mobile devices.

## Edge/cloud inference

*Xu et al. (2023)* proposed a pneumonia detection scheme based on chest X-ray image classification in an edge computing environment, which relieves the computational pressure of centralized data centers. *Shi, Duan & Chen (2021)* deployed the diagnostic network on the edge nodes, which maintained the network training and diagnostic inference services, enabling automated diagnosis, but falling short of optimal resource allocation. *He et al. (2023)* proposed a cloud-based medical image segmentation method using multi-feature extraction and interactive fusion, which uses cloud computing to process a large number of medical images, overcoming the limitations of local computing power. *Kumar et al. (2018)* proposed a new cloud and Internet of Things (IoT)-based neural classifier for detecting and diagnosing serious diseases. The above studies focus on edge or cloud considerations and ignore the impact of energy consumption and latency.

## Collaborative inference

Research related to collaborative inference can be divided into three areas: model partitioning, feature compression, and computational offloading scheduling. Table 1 summarizes the limitations of each and the key features of the MOCI strategy.

### Model partitioning

Collaborative inference splits a DNN into two or more parts and executes each part on a different device. Collaborative inference emphasizes the deployment of DNNs, whereas MEC focuses on the offloading process. The partitioning of DNNs can be performed horizontally or vertically, or a DNN task can be used directly as the unit of partitioning.

    *Kang et al. (2017)* first proposed to partition DNNs for deployment by designing a lightweight scheduler, Neurosurgeon, which can automatically partition DNNs between mobile devices and data centers according to the granularity of neural network layers. *Mao et al. (2017)* proposed MoDNN, a locally distributed mobile computing system for DNN applications, which partitions an already trained DNN model across multiple mobile devices to accelerate DNN computation by reducing device level computational cost and memory consumption. *Zhao, Barijough & Gerstlauer (2018)* proposed DeepThings, a framework for adaptively distributed execution of inference applications on resource-constrained IoT edge clusters, which achieves parallel acceleration by partitioning DNN layers. *Li, Iosifidis & Zhang (2022)* used dynamic programming to solve the fusion block

**Table 1 Comparison of key features of MOCI and other strategies.**

| | Limitation | MOCI |
|---|---|---|
| Model partitioning | • Single test model with poor generalization<br>• Ignore dynamic requirements in complex scenarios | • Compatible with various DNN architectures<br>• Higher generalizability<br>• Optimize partitioning in a dynamic environment |
| Feature compression | • Failure to meet real-time and end-user requirements<br>• Difficulty in balancing compression rate and inference accuracy | • Provide endpoint demand-aware feature compression scheme<br>• Combine with task scheduling to improve system performance |
| Computational offloading and scheduling | • Focus on a single scene and do not analyze the hybrid action space<br>• Difficulty in coping with large-scale tasks and multi-device environments | • Fusion of hybrid action space and multi-agent reinforcement learning techniques<br>• Improve the efficiency of collaborative thinking across multiple devices |

selection problem and used a greedy strategy to achieve edge device selection. However, the above existing studies test a more homogeneous model and do not reflect the generalizability to different DNN architectures.

### Feature compression

The transmission of intermediate features during network model slicing causes a lot of delay and energy consumption, which affects the inference efficiency. Therefore, intermediate feature data must be processed effectively to achieve overall system performance improvement.

*Shao & Zhang (2020)* proposed BottleNet++, an end-to-end architecture consisting of an encoder, a non-trainable channel layer and a decoder, which achieves efficient feature compression and transmission by accounting for the effect of channel noise but ignoring the real-time requirements of the system. *Li et al. (2018)* proposed JALAD, a joint accuracy and latency-aware execution framework that decouples DNNs so that part of the computation is performed on the edge device and the other part is performed in the traditional cloud, with only a minimal amount of data transferred between the two. *Ding et al. (2023)* proposed a new edge cloud DNN collaborative computing framework, JMDC, based on joint modeling and data compression. JMDC uses an attention mechanism to select important model channels for efficient inference and uses quantization techniques to reduce the actual number of bits in the transmitted important intermediate outputs. Although all of the above studies have considered inference between cloud and edge, and decoupled and compressed the model to effectively reduce the inference energy consumption, they have not considered the endpoint requirements and have not further investigated the coordinated assignment of tasks.

### Computational offloading and scheduling

Through computational offloading and scheduling optimization, it achieves a reasonable distribution of DNN components in the MEC environment, thus improving overall system performance.

*Wang et al. (2022)* performed fine-grained decomposition of DNN tasks and proposed a cloud-side-end collaborative dynamic task scheduling mechanism based on DNN layer partitioning technology, which can achieve collaborative computation between the cloud and the side-end of the DNN model and improve the execution efficiency of the DNN model. *Yuan et al. (2023)* considered a continuous DNN inference task and proposed an algorithm based on asynchronous dominance actor-critic (A3C) to manage the transmission power and channel selection of terminals. *Su et al. (2022)* designed a DNN partitioning and heuristic computational resource allocation algorithm based on deep deterministic policy gradient based on Lyapunov optimization technique and reinforcement learning, which effectively reduces the complexity of policy training by observing the environment to train the policy to dynamically decide the DNN partition. *Zhang et al. (2023)* studied the joint energy and delay optimization problem for DNN partitioning and task offloading, and used the PPO-based DPTO algorithm to solve the DNN partitioning and task offloading problem so that the mobile device can make the best scheduling decisions. *Xiao et al. (2022a)* proposed an energy-efficient MEC collaborative inference scheme based on MARL, which allows each mobile device to select both the partition point for deep learning and the collaborative edge for each mobile device based on the number of images, channel conditions, and pre-inference performance. Since the optimal partition point and edge selection depends on the inference cost model of a particular deep learning architecture and the channel model from the device to the edge server, which is challenging in practical MEC, *Xiao et al. (2022b)* also proposed an energy-efficient collaborative inference scheme for MEC based on MARL, which selects the partition point of the deep learning model and the collaborative edge servers based on the environmental conditions. All of the above studies focus on discrete or continuous action spaces, consider a single scenario, and do not analyze and discuss hybrid actions in practical applications.

# SYSTEM OVERVIEW AND PROBLEM MODELING

## System overview

In the task of mobile medical image inference in MEC scenarios, to meet the requirements of DNN inference for high accuracy, low latency and energy consumption, we propose a collaborative inference strategy MOCI. The overall process is illustrated in Fig. 1.

In the MOCI strategy, the processes of task assignment, model partitioning and data transfer are closely coordinated to achieve efficient edge-end collaborative inference. First, the UE acquires image data from the medical imaging device and performs preliminary data processing and feature extraction locally. The UE extracts image features using deep learning models and compresses these features using autoencoder and quantization techniques to reduce the amount of data to be transmitted. The compressed intermediate feature data is then uploaded to the edge server. On the edge server side, the optimal segmentation point of the DNN model is first determined, followed by the partitioning of the model into different parts to perform inference tasks. The edge server processes the intermediate feature data transferred from the UE and performs more complex inference tasks. Finally, the edge server completes the inference and returns the diagnosis results to

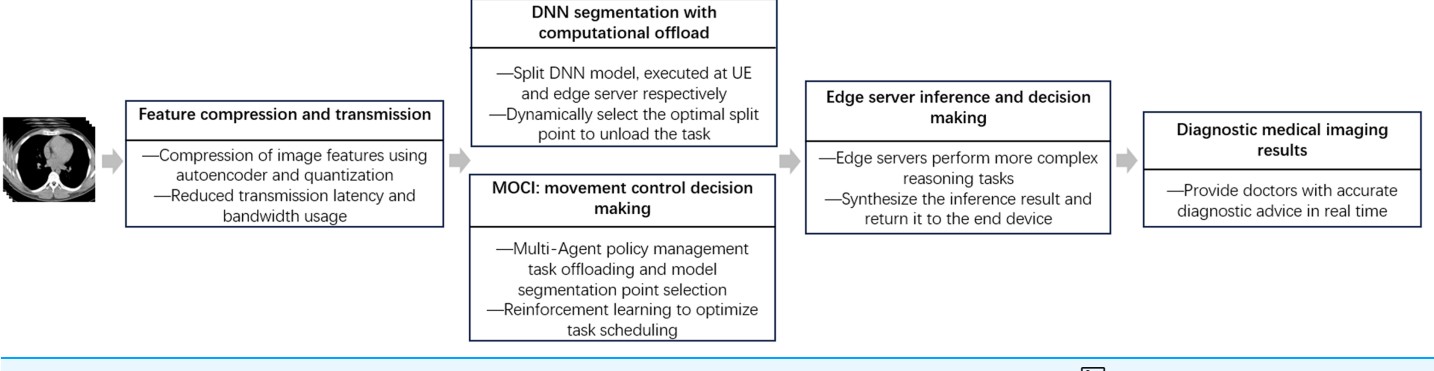

**Figure 1 General flow diagram.**

the UE, which then displays the final inference results for further diagnosis and decision making by the doctor.

In this section, we consider a cooperative end-edge environment consisting of multiple UEs, wireless base stations (BSs), and edge servers. And investigate the energy and delay optimization problem for neural network model partitioning and task offload scheduling in MEC scenarios. In the system, an edge server typically has to serve multiple UEs, *i.e.*, multiple UEs perform model inference for disease classification and diagnosis in medical imaging using only one edge server. The UEs communicate with the BSs *via* a wireless channel, and the BSs are connected to the edge servers *via* optical fiber. The edge server makes a decision based on the current state of the system and decides whether the task should be processed on the UE holding the task or on the edge server. The decision-making process takes into account several factors, including computational resources, communication overhead, and latency energy consumption. The overall goal of the system is to optimize the system's decision-making so that all tasks are completed in the shortest possible time while minimizing the energy consumption of the whole system. The system scenario diagram is shown in Fig. 2.

## Task inference model

The device edge collaborative inference framework splits the neural network on mobile devices and edge servers to decouple the DNN into multiple parts. Taking the deep residual model as an example, the DNN can be divided into layers or residual blocks. The set of end-users is denoted by $U \in \{1, 2, \ldots, N\}$, where $N$ is an integer denoting the total number of UEs in the set. At the terminal, status information for each user includes the number of tasks currently outstanding on the device, the time remaining for local inference and feature compression for tasks currently completed, the size of the remaining data to be offloaded, and the distance from the UE to the edge server.

Both each UE and the edge server use a pre-trained neural network model to compute their tasks. Consider a case where the length of a time slot is $T$. At the end of the previous frame, each UE sends its user state information to the edge server, which collects the states of all UEs and stores them in the state pool.

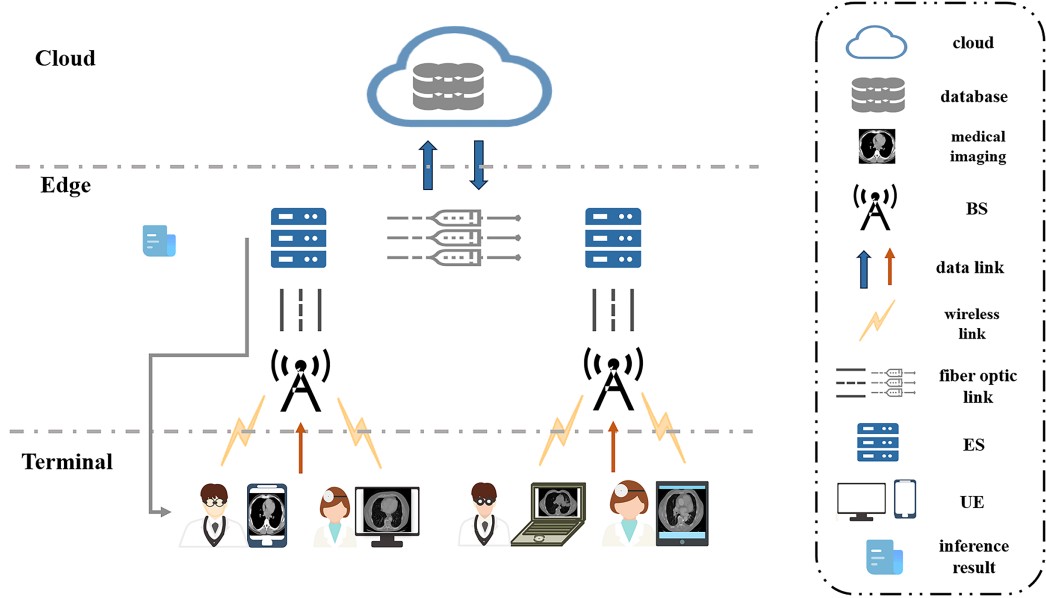

**Figure 2  System scenario diagram.**

In the above environment, each UE has to perform several DNN model inference tasks. The DNN model executed by the UE $U_i$ can be represented as $J_i = \{D, L, \alpha, \beta\}$, where $D$ denotes the dimension of the DNN input layer, each layer of the DNN is denoted as $L = \{l_1, l_2, \ldots\ldots, l_n\}$, and n denotes the number of layers of the DNN. $\alpha$ and $\beta$ represent the sensitivity of the DNN inference task to delay and energy consumption, respectively. In this article, we define division points $d(d \in \{0, 1, \ldots\ldots, H\})$ to divide the model, and the division points determine the layer at which the DNN tasks are divided. It is assumed that the model deployed on terminal $N$ has $H_n$ partition points, which means that the model deployed on $N$ can be divided into $H_n + 1$ parts. If $d = 0$ or $d = H$, it means that there is no partitioning of the DNN model and the whole model is inferred on the edge server or end device. When $d \in [1, H - 1]$, it indicates that the DNN is divided. Layers 1 to $d$ are inferred on the end device, and layers $d + 1$ to $H$ are offloaded to the edge server for inference. The intermediate features output from the inference of layer $d$ are transmitted to the BS through the wireless channel, and then the BS transmits these data to the edge server through the optical fiber, and the edge server, after completing the inference task after layer $H$, transmits the inference results back to the end device.

## Communication model

During the model data offload process, the UE first communicates with the BS *via* a wireless channel. The data is then transmitted from the BS to the edge server *via* a high-speed fiber network, and each user transmits the data at a specific power level *via* a specific offload channel. The offload channel and transmission power of UE $n$ are denoted by $c_n$ and $p_n$, respectively, where $c_n \in \{1, \ldots\ldots, C\}$ and $p_n > 0$. Together with the segmentation point $d_n$, they also form the inference action of UE $n$, denoted as $(c_n, p_n, d_n)$. In the inference action, the split point determines which part of the task is split into local and

offloaded computation. The offload channel determines which wireless channel is used to transmit the data, and the channel selection must take into account the current channel usage and channel quality to ensure the reliability and efficiency of the data transmission. The transmission power determines the amount of power used for the data transmission, and the appropriate transmission power should ensure the success rate of the data transmission and save as much energy as possible. The decision maker in the system provides inference actions for all UEs based on the policy and selects the optimal combination of inference actions for each UE based on the current system state and the policy. A policy is a probability distribution of all possible inference actions that selects the optimal inference action for each UE based on historical data and the current state of the system. Based on the above definition, the uplink data rate between UEs and edge servers can be calculated, and according to Shannon's theorem:

$$R_n(\pi) = b_n \log_2\left(1 + \frac{p_n g_n}{\kappa_n + \sum_{i \in N \setminus \{n\}} p_i g_i}\right) \tag{1}$$

where $b_n$ is the bandwidth of channel $c_n$, $g_n$ is the channel gain between UE $n$ and radio channel $c_n$, and $\kappa_n$ is the background noise power of channel $c_n$.

## Computation model

In the MEC scenario, the latency due to task segmentation and the additional latency incurred on the server in the worst case is a small constant (*Chen & Wang, 2020*), which is negligible compared to the total latency during the processing of the entire diagnostic image classification task and does not affect the optimization results. The total collaborative inference delay in DNN collaborative inference consists of three components: local inference delay $t_n^{local}$, feature compression delay $t_n^{comp}$, and data transmission delay $t_n^{tran}$. The latency and energy consumption of local inference are a result of performing $[1, d]$ layer inference, and the latency and energy consumption of feature compression are mainly incurred during data processing and transmission in edge environments due to the additional time delay and energy consumption caused by compression and decompression of intermediate features and quantization operations. Based on the communication model, both the local inference delay and the feature compression delay can be measured at the device. The data transmission delay is:

$$t_n^{tran} = \frac{S_n^{input}}{R_n(\pi)} \tag{2}$$

where $S_n^{input}$ is the size of the original input sample.

The overall delay can be computed as

$$t_n(\pi) = \begin{cases} t_n^{local}, & d = 0 \\ t_n^M, & d \in [1, \ H-1] \\ t_n^{tran}, & d = H \end{cases} \tag{3}$$

where $t_n^M = t_n^{local} + t_n^{comp} + t_n^{tran}$.

Similarly, we can derive the total energy consumption as

$$e_n(\pi) = \begin{cases} e_n^{local}, & d = 0 \\ e_n^M, & d \in [1, \ H-1] \\ e_n^{tran}, & d = H \end{cases}. \tag{4}$$

where $e_n^M = e_n^{local} + e_n^{comp} + e_n^{tran}$, $e_n^{local}$ and $e_n^{comp}$ are the local inference energy and feature compression energy, respectively, that can be collected on the device, and $e_n^{tran}$ is the data transmission energy:

$$e_n^{tran} = p_n * t_n^{tran}. \tag{5}$$

## Problem formulation

Based on the above-established model, the UE decides the number of DNN layers to be computed at the MEC server and locally based on the current channel situation. Our goal is to find a strategy $\pi$ that minimizes latency and energy consumption for all image diagnosis classification tasks. Assume that the number of tasks received by UE $n$ is $Z_n$ and the number of tasks expected to be completed within a time horizon (*Hao et al., 2022*) is $Z(\pi)$. To fully utilize the communication and computation resources and reduce the overall cost while satisfying the partitioning point, delay and energy consumption constraints, we formulate the optimization of this system as a joint problem of model partitioning and task offload scheduling:

$$(P) \quad V_{n,i} = \alpha \cdot \frac{\max_n \sum_{i=1}^{Z_n} t_{n,i}(\pi)}{Z(\pi)} + \beta \cdot \frac{\sum_{n=1}^{N} \sum_{i=1}^{Z_n} e_{n,i}(\pi)}{Z(\pi)}$$

$$s.t. \ C1: \alpha, \beta \in [0, 1],$$
$$C2: \alpha + \beta = 1,$$
$$C3: d \in \{0, 1, \ldots\ldots, H\}, \quad \forall n \in N$$
$$C4: c_n \in \{1, \ldots\ldots, C\}, \quad \forall n \in N$$
$$C5: 0 < p_n < p_{max}, \quad \forall n \in N$$

$$\tag{6}$$

where $p_{max}$ is the maximum transmit power of a single UE.

The problem is a mixed-integer nonlinear programming problem of the NP-hard class. To address this challenge, we model the whole system as a Markov Decision Process (MDP), define the corresponding state, action and reward spaces within this framework, and use reinforcement learning (RL)-based methods to solve such problems.

MDP is a mathematical framework for modeling decision problems, which describes the process of decision-making by an agent in a stochastic environment. The core of MDP is its ability to systematically describe the state of the environment, the actions of the agent, the probability of state transition, and the rewards. In this section, we define the MDP $M$ as:

$$M = (S, A, P, R, \gamma) \tag{7}$$

where $S$ is the state space, including all possible system states at each time step. $A$ is the action space, including all possible actions that can be taken by the agent in each state. $P$ is the state transfer function, which defines the transfer relationship between a state and an action. $P(s'|s, a)$ denotes the probability of transitioning to $s'$ after performing an action $a$ when the state is $s$. $R$ is the reward function, which represents the reward obtained by transitioning from one state to another reward obtained by acting to move to another state. $\gamma \in [0, 1]$ is the discount factor used to weigh the relative importance of immediate rewards against potential future rewards. In the defined MDP framework, the goal of an agent is to learn to find an optimal task offload scheduling strategy by interacting with the environment, which is capable of selecting the most appropriate action in a series of state-action decision processes to maximize cumulative discount rewards.

For the three core components of MDP—state space, action space, and reward function—the detailed definitions are as follows:

(1) State space: In the $t$th decision time frame, the state space includes the number of remaining tasks $k$, the remaining local computation time $l$, the remaining offloaded data size $n$, and the distance $h$ from the UE to the edge server, *i.e.*,

$$s = (k, l, n, h) \tag{8}$$

In collaborative inference, the state includes the current task progress and resource usage of all user equipment, such as the tasks being processed by each UE, allocated resources, channel selection and current transmit power.

(2) Action space: after obtaining the current state, the action at time t consists of the offload channel $c$, the transmit power $p$, and the partition point $d$, *i.e.*,

$$a = (c, p, d) \tag{9}$$

In a multi-user environment, action $a$ allows the system to dynamically adjust the resource allocation policy according to the current at each decision time frame, which can effectively balance computational and transmission loads to better cope with ever-changing task demands and network conditions.

(3) Reward function: After the agent has performed the current action, the environment feeds back the corresponding reward value, which is related to the original optimization problem described in problem $(P)$. Since the expectation $Z(\pi)$ in problem $(P)$ is difficult to obtain, the number of completed tasks $Z_t$ at time $t$ is used as an estimate of $Z(\pi)$, and a similar definition is proved in *Hao et al. (2022)*. In this article, we aim to minimize inference latency and energy consumption, *i.e.*, minimize the integrated cost of inference, and set a negative value of the integrated cost as the reward value:

$$r_t = -V_{n,i} = -\alpha \cdot \frac{T_0}{Z_t} - \beta \cdot \frac{E_t}{Z_t} \tag{10}$$

$$s.t. \quad C1 - C5$$

where $T_0$ is the duration of a time frame and $E_t$ is the energy consumption at time $t$.

# SYSTEM DESIGN

## Feature compression

Medical images contain complex anatomical structures and pathological features that require fine feature extraction and analysis. Most medical images (*e.g.*, computer tomography (CT) and magnetic resonance imaging (MRI)) are mainly grey-scale images rather than color images, and intensity values in grey-scale images indicate different tissue densities or compositions. Due to these characteristics, common image processing methods are often not effective enough on medical image data and require the use of specialized feature compression methods and techniques. Medical images require high-precision reconstruction to ensure diagnostic accuracy, and multi-layer coding decoders based on convolutional neural networks excel in feature extraction and reconstruction. For chest CT images, feature compression and noise reduction using a multi-layer coding decoder can help to extract clear lung, vessel, and airway structures and improve the accuracy of lesion detection and diagnosis.

Meanwhile, existing compressors in cloud-side end environments usually require large overheads and incur high latency. To achieve efficient intermediate feature compression, we propose a multi-layer autoencoder-based compression method that can better capture image features by using regularization to prevent overfitting and combining autoencoder and quantization modules to compress intermediate features. An autoencoder is an unsupervised learning model commonly used for tasks such as feature extraction, dimensionality reduction, data compression and noise reduction. It consists of an encoder, which compresses the input data into a low-dimensional latent representation, and a decoder, which reconstructs the original input from this latent representation. Figure 3 illustrates the collaborative inference workflow using the feature compression approach.

The encoder gradually reduces the number of channels of the input feature map through a series of convolutional layers that use ReLU activation functions and batch normalization to improve the training effectiveness and stability of the network and preserve the important image feature information. The decoder gradually restores the number of channels of the compressed feature map to the number of channels of the original input through a series of convolutional layers, and this structural design allows the model to maintain high reconstruction accuracy and feature expressiveness when processing unseen image data. The shape of the intermediate features is assumed to be $(C, W, H)$, where each dimension represents the number of tensor channels, width, and height (*Li et al., 2024*). The encoder is placed at the rear of the edge device network and receives intermediate features of size $C \times W \times H$. After a series of convolution and activation operations, the resulting compressed features have a size of $C' \times W' \times H'$, where $C' < C$.

When the model is used for UE services, the input data may deviate from the distribution of the original training data set, resulting in a degradation of the model performance. Traditional feature compression methods tend to suffer from significant performance degradation in the face of changes in the input data distribution. In contrast,

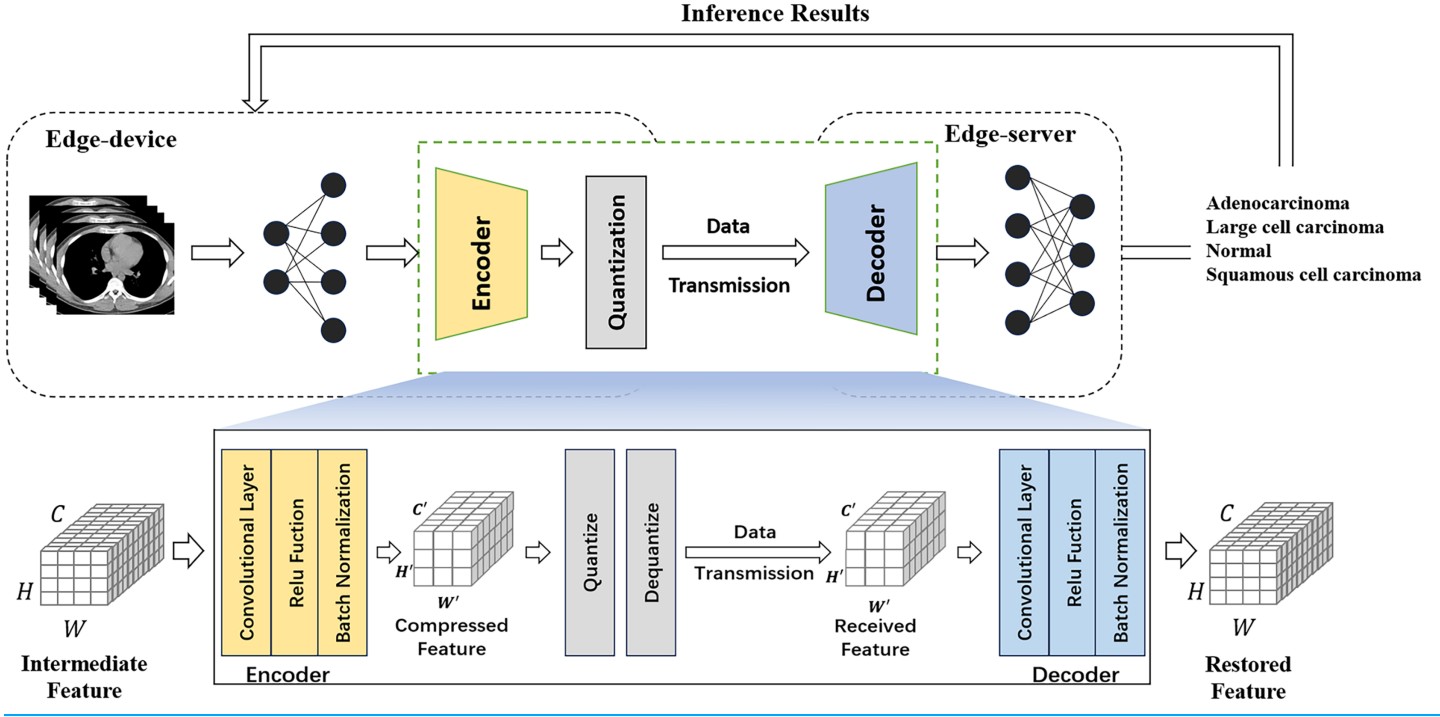

**Figure 3 Feature compression flowchart.**

the introduction of the batch normalization layer further enhances the adaptability of the model to changes in the input data distribution, enabling the proposed lightweight autoencoder to cope with such situations more effectively.

Feature compression for medical image data in edge environments should not only ensure low latency and low power consumption but also tightly control its accuracy to provide physicians with reliable diagnoses. Existing work has shown that using lower bit-width representations of intermediate features has little impact on inference accuracy (*Li et al., 2018*), so quantization techniques are used to further compress the output features of the encoder. Quantization is an important method for deep learning model compression, the core idea being to convert the floating-point parameters in the model to low precision integer values to reduce the storage and computational overhead of the model, and to help optimize the performance and resource consumption of the model on edge devices. Dequantization is the process of converting the quantized integer values back to floating point numbers. The quantization technique used in our experiments is uniform quantization, which refers to the mapping of floating-point values to a uniformly distributed range of integers. It assumes that the distribution of the original floating-point data is uniform and uses a fixed step size to map to discrete integer values.

On the UE, the quantization procedure can be formulated as

$$q(x) = round\left(\frac{x - min}{\Delta}\right) \qquad (11)$$

where $\Delta = \frac{max - min}{2^{qbit} - 1}$ is the quantization step size, which is used to indicate the size of the quantization interval. $x$ is the intermediate feature to be quantized. $max,\ min$ are the maximum and minimum values of the intermediate features to be quantized, which can be replaced by the result computed on a pre-collected set of feature maps. $qbit$ is the bit width used for quantization and $q(x)$ is the output integer of $x$. On the edge server, the quantized value can be recovered approximately by the dequantization procedure:

$$x' = q(x) * \Delta + min \tag{12}$$

where $x'$ is the recovery value. As rounding operations in quantization introduce rounding errors, $x'$ is not usually exactly equivalent to $x$.

The overall compression rate of our proposed compression technique, which includes coding, decoding and quantization, is given by:

$$R = \frac{C \times b}{C' \times qbit} \tag{13}$$

where $b$ is the original intermediate feature before quantization, represented by a 32-bit floating point number. $b/qbit$ indicates the compression rate of the quantization process.

For the pre-trained model trained in this stage and the selected segmentation points, the autoencoder is first considered to be trained by the distance between the original features and the recovered features, and the cross-entropy loss between the predicted outputs and the real labels is also introduced to minimize the prediction error and improve the prediction accuracy. The loss function for training the autoencoder is formulated as:

$$Loss = \left|\left|F_i^x - F_o^x\right|\right|_2 + \eta L_{ce}(f(x), y) \tag{14}$$

where $f$ is the pre-training model, $x$ is the training sample, $F_i^x, F_o^x$ denote the original and recovered features of the autoencoder, respectively, $L_{ce}$ is the cross-entropy metric, $y$ is the true label corresponding to the sample $x$, and $\eta$ is the hyperparameter that compensates for the losses of the above two components.

## DNN segmentation scheduling strategy

The proposed MOCI strategy, based on multi-agent proximal policy optimization (MAPPO), solves the collaborative inference scheduling problem in the medical image classification task for edge scenes, and the overall workflow is shown in Fig. 4. The multi-intelligentsia collaborative inference problem contains a mixture of action spaces, such as discrete off-load channels and continuous transmit power. The overall optimization objective is to maximize the total inference reward for all DNN tasks:

$$max \sum_{t=1}^{T(\pi)} r_t \tag{15}$$

where $T(\pi)$ is the total time to complete all tasks.

In traditional reinforcement learning, an agent obtains rewards by interacting with its environment and maximizes the cumulative rewards by optimizing its policy. PPO is a

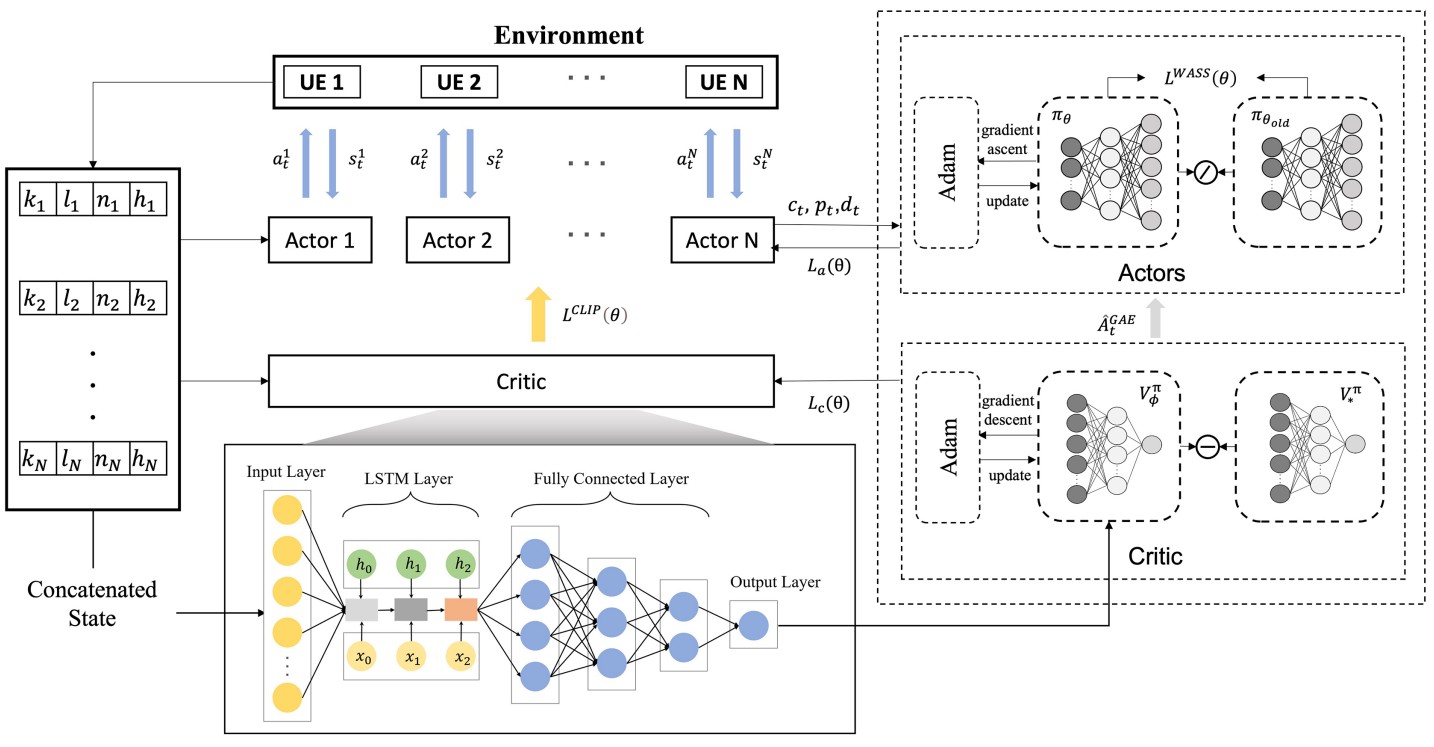

**Figure 4** Scheduling flowchart for collaborative inference.    

policy-based reinforcement learning algorithm, a variant of the Actor-Critic framework (*Gu et al., 2021*). It can deal with both discrete and continuous action spaces and aims to maximize its long-term rewards by updating the policy so that the agent chooses the optimal action in a given state.

MAPPO is an extension of the PPO algorithm in a MARL environment. Similar to single-agent PPO, MAPPO also uses the Actor-Critic framework. The input of the actor network is the state, and the output is the action probability (for discrete action space) or the action probability distribution function (for continuous action space). The input of the critic network is the state, and the output is the value of the state. However, it is adapted to the problem of synergy and competition among multiple agents, each of which has its own strategy and updates and optimizes its behavioral strategy by interacting with other agents. To achieve collaboration between agent, we use a critic network, which contains global information, and an actor network, which contains local information.

In a MARL environment where collaborative inference is performed, the main task of the actor network is to output a policy $\pi_\theta(a_t|s_t)$, *i.e.*, to predict the distribution of actions $a_t$ given a state $s_t$. Each branch of the actor network generates different types of actions, respectively. For discrete actions, each branch of the actor network outputs the probabilities of the different possible actions, $p_i(s_t)$, through a softmax function. The softmax function ensures that these probabilities sum to 1, thus forming a valid probability distribution. The choice of discrete actions follows a categorical distribution.

$$\pi_{\theta_n}^d \left( a_{t,n}^d | s_t \right) = \prod_{i=1}^{A} p_i(s_t) I_{\left\{ a_{t,n}^d = i \right\}}, \ \forall n \in N \tag{16}$$

where $\pi_{\theta_n}^d$ denotes the distribution of strategies in the discrete action part, $A$ is the number of possible actions and the indicator function $I$ takes the value 1 if $a_{t,n}^d = i$ and 0 otherwise. Actor networks are capable of generating reasonable action distributions in the current state, leading to effective decision-making and cooperation in complex multi-agent environments.

For continuous actions, the corresponding branch of the actor network outputs the mean $\mu(s_t)$ and standard deviation $\sigma(s_t)$ of the action, which are calculated based on the current state $s_t$. The successive actions are assumed to follow a normal distribution, *i.e.*,

$$\pi_{\theta_n}^c \left( a_{t,n}^c | s_t \right) \sim N(\mu(s_t), \sigma^2(s_t)), \ \forall n \in N \tag{17}$$

where $\pi_{\theta_n}^c$ denotes the policy distribution of the continuous action component. The actor network can generate a distribution of continuous actions in a given state and determine specific action values by sampling from a normal distribution.

In this article, the scenario contains multiple UEs and uses multiple actor networks to provide inference decisions for the UEs, *i.e.*, each UE has an independent actor network, and these networks process the input state information by sharing some pre-layers and generating different decisions respectively, with the number of actor networks equal to the number of UEs. To cope with the challenges of the hybrid action space, each actor network has three output branches, which are responsible for deciding the task partition point, the data offload channel, and the transmission power, respectively. The processing results of the state information are shared by the pre-layers, enabling the system to use this information more efficiently to effectively deal with the hybrid action space and enhance the decision-making and collaboration capabilities of the multi-agent system. Each UE can make decisions independently while sharing state information to improve overall performance. The algorithmic pseudo-code for the MOCI strategy is shown in Fig. 5.

The original strategy gradient algorithm is very sensitive to step size, but it is difficult to choose an appropriate step size, which is detrimental if the difference between the old and new strategy changes during training is too large. PPO proposes a new objective function that can be updated in small batches over multiple training steps, which solves the problem of difficult determination in the strategy gradient algorithm. Although a clipping function based on the probability ratio of the old and new strategies is used in the PPO algorithm to constrain the difference between the old and new strategies, recent studies have shown that this does not strictly constrain the difference between the old and new strategies (*Cheng, Huang & Wang, 2021*). Therefore, the MOCI strategy adds Wasserstein distance to the original strategy to further constrain it. Meanwhile, to effectively deal with the dynamic network conditions and task complexity, the LSTM structure is combined with the critic network. LSTM is a special type of recurrent neural network (RNN) designed to solve the gradient vanishing and gradient explosion problems of traditional RNNs when dealing with long time series data. LSTM can store long-term dependencies in sequences by

---

**Algorithm 1** MOCI algorithm

1: Initialize replay memory $D$, system environment and initial state $s_0$
2: Randomly initialize parameters of the actors and the critic as $\theta_n$ and $\phi$, $n \in N$
3: New state $s_t \leftarrow s_0$
4: **for** timestep $t := 1$ to $T_{max}$ **do**
5:     **while** $D$ is not filled **do**
6:         Choose action $a_t$ from current strategy $\pi_\theta(a_t|s_t)$
7:         Execute $a_t$ and receive feedback from the environment: reward $r_t$, the next state $s_{t+1}$ and related information
8:         Store tuple $(s_t, a_t, r_t, s_{t+1})$ into $D$
9:         Compute delay, energy consumption via (2)-(5)
10:        **if** $s_{t+1}$ is not the terminal state **then**
11:            $s_t \leftarrow s_{t+1}$
12:            $t \leftarrow t+1$
13:        **else**
14:            Reinitialize the current state $s_t$ to $s_{t+1}$
15:        **end if**
16:     **end while**
17:     Compute cumulative rewards for status values via (15)
18:     Obtain the generalized advantage estimation via (18)
19:     $I_{max} \leftarrow$ The number of complete batches in the current buffer
20:     **for** iterations $i := 1$ to $I_{max}$ **do**
21:         Randomly sample a minibatch $B$ from $D$
22:         Apply LSTM to enhance critic network
23:         Compute $L_c(\phi)$ with $B$ via (19)
24:         Compute and $L_a^{CLIP}(\theta)$ with $B$ via (20)
25:         Update critic: $\phi \leftarrow \phi - \alpha \nabla_\phi L_c(\phi)$
26:         **for** $n := 1$ to $N$ **do**
27:            Compute $L^{WASS}(\theta)$ via (23)
28:            Compute $L_a(\theta)$ via (24)
29:            Update actor: $\theta_n \leftarrow \theta_n - \alpha \nabla_{\theta_n} L_c(\theta)$
30:         **end for**
31:     **end for**
32:     Clear memories in $D$
33: **end for**

---

**Figure 5** The algorithmic pseudo-code for the MOCI strategy.

introducing a structure called a "gating mechanism" and effectively avoids the problems of traditional RNNs in learning long sequences. In MOCI, where the task load changes dynamically, the LSTM is able to maintain long-term memory through its memory cells, remembering historical inputs and transferring this information to future time steps. We exploit the processing power of LSTM on sequence data to improve the accuracy and stability of the critic network in evaluating the state-action pairs.

When training, we first need to calculate the dominance score for each experience based on the data from the experience pool; this score reflects the degree of superiority or inferiority of each action in the same state. The state score is the average of the action scores, while the dominance score is the difference between the action score and the state score. A higher dominance value indicates a higher return that the action can bring, and in the proposed framework the dominance value is calculated using the generalized advantage estimation (GAE) algorithm (*Schulman et al., 2015*):

$$\hat{A}_t^{GAE(\gamma,\lambda)} = \sum_{t'=t}^{T(\pi)} (\gamma\lambda)^{t'-t} \left( r_t + \gamma V_\phi^\pi(s_{t+1}) - \sum_{t'=t}^{T(\pi)} \gamma^{t'-t} r_{t'} \right) \qquad (18)$$

where $\lambda \in [0, 1]$ is a hyperparameter controlling the degree of smoothing of the dominance estimate. $t$ is the current time, $V$ is the state value function, $s_{t+1}$ is the state at time $t + 1$, and $r_{t'}$ is the reward at time $t'$.

Simplify $\sum_{t'=t}^{T(\Pi)} \gamma^{t'-t} r_{t'}$ to $V_*^\Pi(s_t)$, which denotes the true cumulative reward of state $s_t$ in each time frame. Then, for the critic network, we can formulate the loss function as:

$$L(\phi) = \left\| V_\phi^\Pi(s_t) - V_*^\Pi(s_t) \right\|_2 \qquad (19)$$

where $\phi$ is a model parameter and $s_t$ is the state at time $t$.

For the actor network, training is performed using a loss clipping strategy, where the clipped loss can be formulated as:

$$L^{CLIP}(\theta) = \hat{E}_t \left[ \min\left( r_t(\theta)\hat{A}_t^{GAE(\gamma,\lambda)}, clip(r_t(\theta), 1-\epsilon, 1+\epsilon)\hat{A}_t^{GAE(\gamma,\lambda)} \right) \right] \qquad (20)$$

where $\theta$ is a model parameter, $\hat{E}_t$ is the expectation of the objective function at time $t$, $\epsilon$ is a trimming factor to control the magnitude of the strategy update, $\pi_{\theta_{old}}$ is the strategy function before the update, and $\pi_\theta$ is the strategy function after the update. $r_t(\theta)$ is the ratio of the old and new strategies, denoted as

$$r_t(\theta) = \frac{\pi_\theta(a_t|s_t)}{\pi_{\theta_{old}}(a_t|s_t)} \qquad (21)$$

where $a_t$ is the action performed at time $t$.

Furthermore, the clip function restricts $r_t(\theta)$ to $[1 - \epsilon, 1 + \epsilon]$ to avoid too large policy updates, which is computed as

$$clip(r_t(\theta), 1-\epsilon, 1+\epsilon) = \begin{cases} 1-\epsilon, & r_t(\theta) < 1-\epsilon \\ 1+\epsilon, & r_t(\theta) \geq 1+\epsilon \end{cases} \qquad (22)$$

The algorithm constrains the magnitude of the strategy update by the old-to-new strategy ratio $r_t(\theta)$. Although the clipping strategy based on the old-to-new strategy probability ratio can constrain the strategy update within a local range (*i.e.*, within the range $[1 - \epsilon, 1 + \epsilon]$), the constraining effect of the clipping strategy is weakened once it goes beyond this range. Since the clipping strategy mainly focuses on the change in the

probability ratio rather than the change in the overall strategy distribution, even if the clipping strategy works, the difference between the overall distributions of the old and new strategies may still be large. Therefore, we introduce the Wasserstein distance to constrain the old and new strategies more tightly. The Wasserstein distance is a measure of the difference between two probability distributions, the minimum 'transport cost' required to convert one probability distribution into another that more accurately reflects the difference between the two distributions. The loss function for the Wasserstein distance is given by

$$L^{WASS}(\theta) = W(\pi_\theta, \pi_{\theta_{old}}) = inf_{\chi \in \Pi(a_t, s_t)} E_{(\pi_\theta, \pi_{\theta_{old}}) \sim \chi}[||\pi_\theta, \pi_{\theta_{old}}||] \tag{23}$$

where denotes all joint distributions $\chi(\pi_\theta, \pi_{\theta_{old}})$, $E$ denotes the expected value with respect to the joint distribution $\chi$, and $||\pi_\theta - \pi_{\theta_{old}}||$ denotes the Wasserstein distance.

Therefore, the objective function of the actor network is calculated as:

$$L(\theta) = \sum_{n=1}^{N} \left[ L^{CLIP}(\theta) + \zeta L^{WASS}(\theta) + \xi E_t[\mathcal{H}(\pi_{\theta_n}(a_t|s_t))] \right] \tag{24}$$

where $\zeta$ and $\xi$ are equilibrium hyperparameters and $\mathcal{H}(\pi_{\theta_n}(a_t|s_t))$ is the entropy reward value, which is used to encourage the agent to explore more action space and make the strategy more diversified.

## EXPERIMENTS

This section demonstrates the compression performance of the proposed compressor, the convergence performance of the MOCI strategy, and the parameter selection for the experiments, and investigates how it effectively reduces the multi-agent collaborative inference overhead of the ResNet18 model. Finally, the proposed MOCI strategy framework is evaluated on two other popular DNN models: VGG11 and MobileNetV2.

### Simulation setup

- **Device configuration:** The device used for the inference process is an Intel(R) Core(TM) i5-7500 CPU @3.40 GHz and an NVIDIA T4 GPU. We use PyCharm as the development tool for the Python IDE. To measure the power consumption, the power consumption is calculated by taking the CPU usage and time in each time interval. Assuming that the power consumption of the system at 100% CPU usage is *pmax*, the power consumption at *cpu_percent*% usage is $p = pmax * (cpu\_percent / 100)$.

- **Dataset:** The dataset used for the experiment is Chest CT-Scan images Dataset (https://tianchi.aliyun.com/dataset/93929), which contains a normal category and three types of chest cancer: adenocarcinoma, large cell carcinoma, and squamous cell carcinoma. The number of samples in each class varies from 100 to 350. 80% of each class was randomly selected as the training set and the remaining samples were used as the test set.

- **Experimental models:** Three DNN models are used for experiments in this section: the ResNet18 (*He et al., 2016*), VGG11 (*Simonyan & Zisserman, 2014*) and MobileNetV2 (*Sandler et al., 2018*). These three models are widely used in the field of deep learning and

have been verified for their performance and reliability by a large number of benchmark tests. ResNet18, known for its residuals, is suitable for scenarios where accuracy is required but computational resources allow; VGG11 is suitable for scenarios where strong feature representation is required; and MobileNetV2 is suitable for resource-constrained edge devices due to its light weight and high efficiency. By using these three models with different characteristics, different application scenarios and device requirements can be covered, and it is possible to comprehensively evaluate the applicability and generalization of the proposed methods in different network architectures.

- **Parameter setting:** In the experiment, five UEs and two communication channels are set up for data transmission. The distance between each UE and the edge server follows the normal distribution $h \sim U(1, 100)$, and each UE receives a certain number of tasks $k$ at the beginning, and $k$ follows the Poisson distribution $Pois(\lambda)$ with parameter 200. When the tasks of all UEs are completed, the current round ends and the parameters $h$ and $k$ are reinitialized at the beginning of a new round, simulating the randomness and uncertainty in the real network environment. The collaborative inference system we have designed is assumed to be located in an urban cellular radio environment (*Rappaport, 2024*), and the channel gain $g_n$ is denoted as $d^{-l}$, where the path loss exponent $l$ is set to 3. For the parameters in the experiments, the best-performing parameters were selected from several candidates through a series of test experiments, while also following the common configurations used in PPO implementations, as shown in Table 2, and some of the parameter test experiments are listed in the section 'Parametric analysis'. For each actor network consisting of fully connected layers, the first two layers are shared layers, containing 256 and 128 neurons respectively. Each output branch also has two layers, the first containing 64 neurons, and the structure of the last layer depends on the type and dimension of the corresponding action. The shared-critical network consists of an LSTM whose outputs are mapped to state values by a linear layer.

- **Performance benchmarks:** To evaluate and validate the performance of the proposed MOCI strategy, this section compares it with the following five algorithms:

1) Local offloading (LO): the entire DNN model is executed on the user's device without segmentation and offloading.

2) Edge offloading (EO): the DNN model is not segmented, everything is offloaded to the edge server for inference, and the edge server returns the inference results to the end device.

3) Random offloading scheduling (RO): a random selection of DNN model segmentation points and a random decision to execute them on terminals or edge servers.

4) Dueling DQN: Joint optimization of DNNs for segmentation and offload scheduling using the Dueling DQN algorithm. Dueling DQN is a DQN-based reinforcement learning method that improves learning efficiency and policy stability by decomposing the Q-value function into state and dominance values.

**Table 2 The value for simulation parameters.**

| Parameters | Value |
| --- | --- |
| Bandwidth per channel $b_n$ | 1 MHz |
| Background noise power $\kappa_n$ | $10^{-9}$ W |
| Discount factor $\gamma$ | 0.95 |
| Duration of each period $T_0$ | 0.5 s |
| Actor learning rate $lr\_a$ | 0.0001 |
| Critic learning rate $lr\_c$ | 0.0001 |
| Memory buffer size | 1,024 |
| Batch size | 256 |
| Reuse time | 20 |
| $\lambda$ | 0.95 |
| $\epsilon$ | 0.2 |
| $\zeta$ | 0.1 |
| $\xi$ | 0.001 |

5) MAPPO: Joint optimization of DNNs for segmentation and offload scheduling using the MAPPO algorithm. MAPPO is a MARL algorithm that is an extension of the single-agent PPO algorithm that allows multiple agents to share information during the policy update process for more efficient collaborative learning.

## Feature compression performance

The experiments used the ResNet18 network as the base model and selected four segmentation points; specifically, a ResNet18 model divides a sample processing into four stages (input processing stage, convolutional feature extraction stage, pooling stage, and fully connected layer stage). The second output layer (batch normalization layer) of each stage is used as the dividing point, resulting in a total of four selected points (*Hao et al., 2022*).

Using point 1 of ResNet18 as an example, we select the optimal compression rate by comparing the accuracy of models trained with different compression rates to the accuracy of the model trained in the local environment. In order to evaluate the impact of different compression rates on the model performance, three compression rates of 48%, 64% and 96% were chosen to measure the accuracy of the model based on a combination of previous research and experimental exploration, representing low, medium and high compression, respectively. Figure 6A illustrates the accuracy achieved by the classification model at different compression rates, and a comparison with the accuracy of the local model. A similar experiment to determine the accuracy rates that can be achieved after compression at each point for the other models results in Fig. 6B. Figure 6B illustrates the maximum compression rate that can be achieved by choosing the closest local model accuracy and within a 2% loss in accuracy, as a 2% loss in accuracy is negligible for most real-world applications when compared to the reduced latency and energy consumption. A

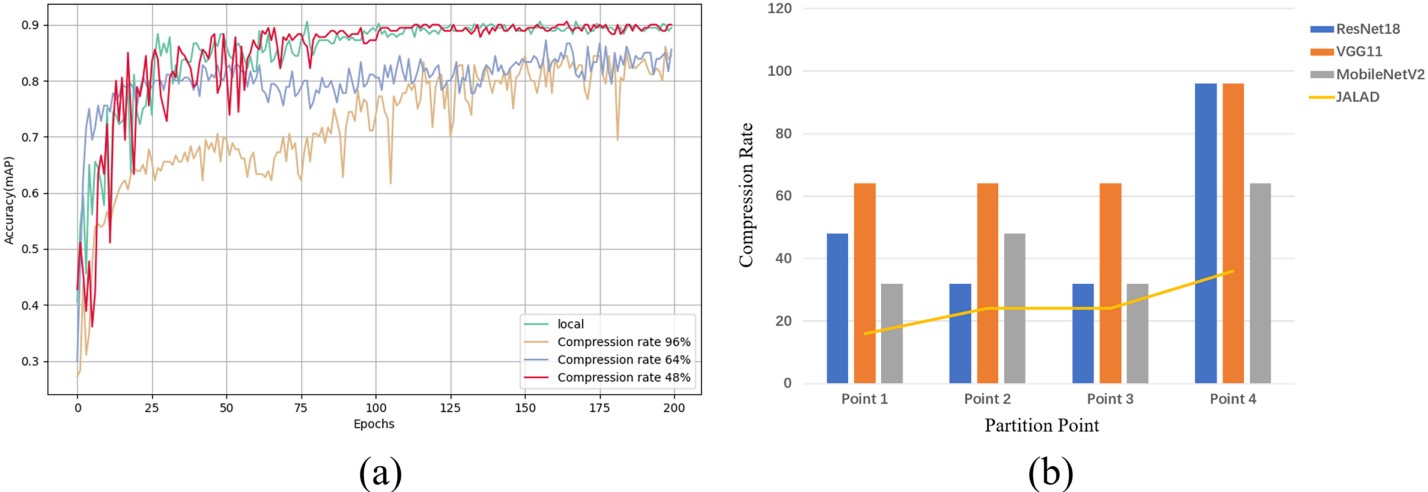

**Figure 6 Maximum achievable compression rate at each point in the range of accuracy loss.** (A) Local model accuracy, and comparison of model accuracy at different compression rates in edge collaboration. (B) Maximum compression rate at each point of the model.

comparison with some other state-of-the-art compression methods is also shown in Fig. 6B, which shows that all three models outperform the other methods at all points.

Table S1 shows the results of ResNet18, VGG11 and MobileNetV2 on the average compression rate after coded quantization, ResNet18 has relatively low compression effect due to the deeper network and residual links, coded quantization can reduce the number of parameters, but the complex structure limits the compression rate; VGG11 has a simple structure, many redundant parameters, coded quantization is easy to VGG11 has a simple structure with many redundant parameters, it is easy to remove the redundant layers and weights when coding and quantizing, and the fully connected layer and convolutional layer are easy to compress and do not easily affect the accuracy; MobileNetV2 is highly optimized, and the deep separable convolution reduces the computation and the number of parameters, and the space for compression is limited.

According to the quantitative analysis in our experiments, the use of the proposed compression quantization method for each model in the edge scenario can achieve a high compression rate with guaranteed accuracy, reduce transmission delay and energy consumption, and thus maximize the cumulative rewards of subsequent offloading scheduling.

## MOCI strategy evaluation

After obtaining the local inference and intermediate feature compression overheads, overall scheduling of the inference model is required to ensure the stability of the interactions between agent and to maximize the cumulative rewards, thus minimizing the overall system latency and energy consumption. The proposed MOCI strategy aims to solve the delay and energy consumption problems of the system (*P*). First, its convergence as well as delay and energy consumption are evaluated, then the effects of different

parameter settings on the system performance are compared. Finally, generalization experiments are performed on other popular network models.

### Convergence assessment

This section analyses the convergence of the MOCI strategy with two other reinforcement learning algorithms (Dueling DQN and MAPPO). Figure 7 shows that all three methods converge, with the MOCI strategy performing best. As the number of iterations increases, the reward values show a faster convergence trend, with the reward of the MOCI strategy growing rapidly during the iteration phase of about the first 30K rounds, followed by a gradual and steady increase with small oscillations. At about the 200Kth iteration, the reward of the MOCI strategy converges to about 0.8. In comparison to the other two algorithms, both the convergence speed and the stability of the training are optimal, which greatly saves the time cost of decision-making and can achieve a lower total energy consumption, indicating that the interoperability between multi-terminal devices and feature compression play a role in the overall planning of the system. The MOCI strategy constrains the old and new strategies through intermediate feature compression as well as LSTM and Wasserstein distance, which significantly improves training performance and convergence stability, and enhances the exploratory capability and global optimization effect during the training process.

### Optimized latency and power assessment

Table S2 shows through the differences in action selection of strategies and cumulative rewards, that the Wasserstein distance significantly reduces the variability between old and new strategies, avoiding over-exploration and instability due to drastic changes in strategies. The data show that with the introduction of the Wasserstein distance constraint, the variability between strategies is smoother and more consistent across multiple trials, allowing higher cumulative rewards to be achieved with fewer training steps.

Table S3 shows the average delay, average energy consumption and reward for the three cases without LSTM, with LSTM and LSTM combined with Wasserstein distance. The data show that when the LSTM structure is introduced, the delay and energy consumption are significantly reduced and the reward value is increased, while the experimental results are further optimized by adding the Wasserstein distance constraints.

The results of evaluating the delay (Fig. 8A) and energy consumption (Fig. 8B) of each strategy as the number of UEs increases are shown in Fig. 8. The experimental results show that as the number of UEs increases, the average inference delay and energy consumption of each strategy, except for LO inference, increase as the available channel resources are fixed. LO does not offload tasks to edge servers and does not rely on wireless communication resources, so the average inference overhead remains constant as the number of UEs varies. The task complexity as well as scheduling difficulty increases with the increase of UE, which leads to the rise of data transmission time and computation/ transmission energy consumption, and consequently, the average delay and average energy consumption of EO, RO, Dueling DQN, MAPPO algorithms as well as the MOCI strategy show an increasing trend. The stochastic nature of RO scheduling lacks the ability to adapt

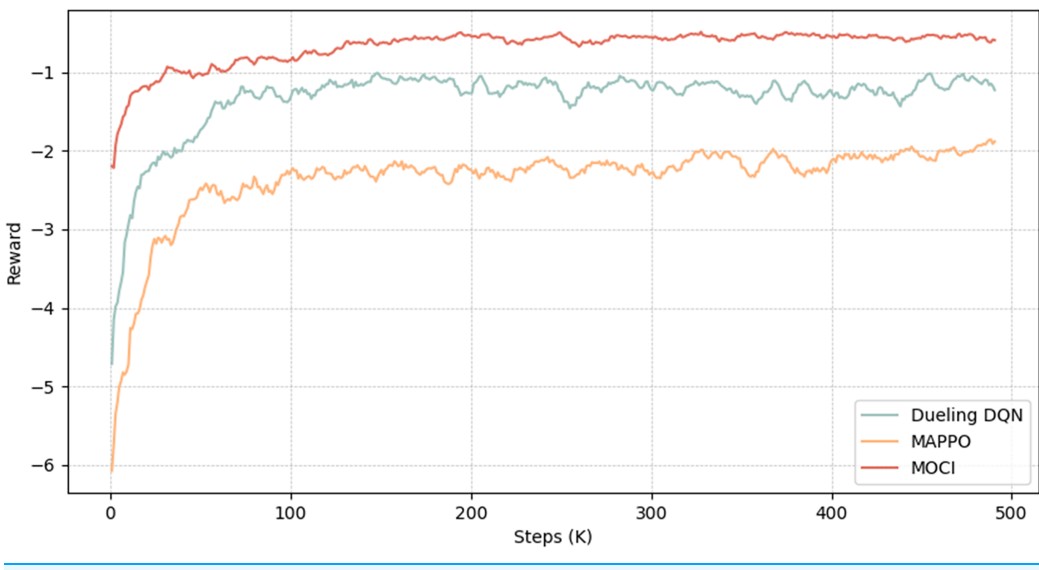

**Figure 7** **Convergence process diagram.**

to changes in the environment, and the Dueling DQN and MAPPO algorithms have a certain computational burden when dealing with complex scheduling in high-dimensional state spaces and hybrid action spaces, resulting in inefficient inference. The MOCI strategy shows more significant advantages over these algorithms, while maintaining a consistently low inference overhead. MOCI achieves lower average latency and has lower average energy consumption regardless of the number of users. Even with a small number of users (*e.g.*, $N = 3$), MOCI maintains optimal performance.

*Parametric analysis*

- $\alpha - \beta$: Figure S1 illustrates the performance of the system's average delay and energy consumption as the parameter $\alpha$, which is the parameter that limits the delay in the optimization objective. The changes in delay and energy consumption show a negative correlation, as $\alpha$ continues to increase, the system's delay requirement increases and the energy consumption requirement gradually decreases, which is more suitable for some delay-sensitive applications such as medical diagnosis, emergency rescue and other areas. If $\alpha = 1$, this means that the optimization objective of the system at this time is to minimize the total delay of the system, without considering the impact of energy consumption. Since $\alpha + \beta = 1$, as $\alpha$ decreases, the parameter $\beta$ that limits energy consumption gradually increases, indicating that the system is strengthening the limitation of energy consumption in the hope of reducing the energy consumption of the system as much as possible, which is more appropriate for some computationally intensive programs, such as AI inference and big data analysis. $\beta = 1$ indicates that the optimization objective of the system at this point is to minimize the total energy consumption of the system band, without considering the impact caused by latency. In contrast, for MEC applications that require both fast response and energy savings (*e.g.*, intelligent video surveillance), a balance between $\alpha$ and $\beta$ needs to be realized. Therefore,

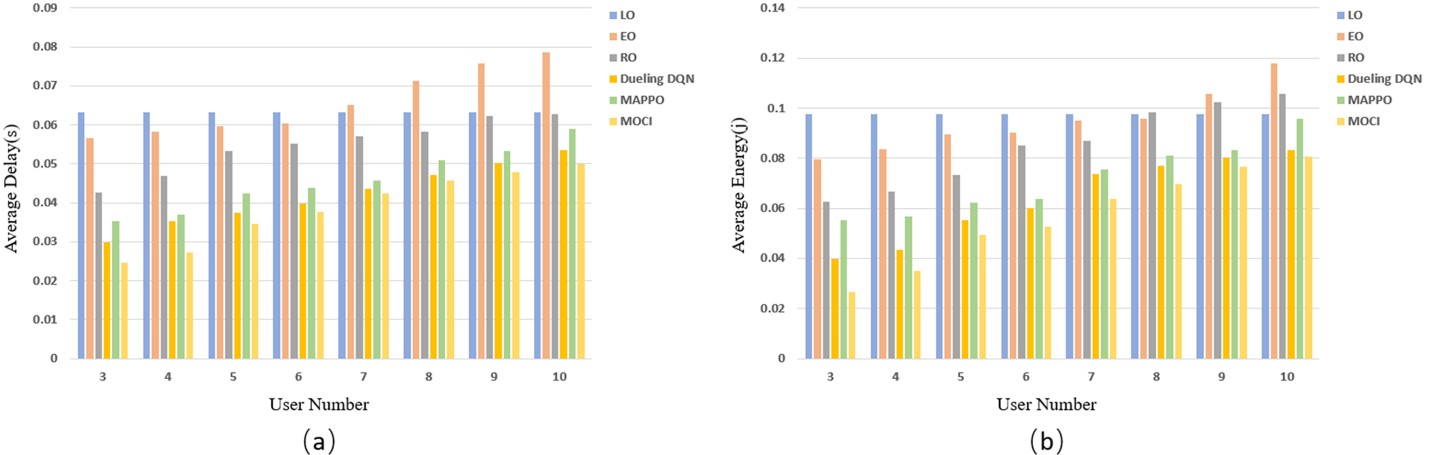

**Figure 8  Comparison of policy delay and energy consumption for different UEs.** (A) Relationship between the number of UE and latency. (B) Relationship between the number of UEs and energy consumption.

a demand-specific balance between delay and energy consumption for different types of MEC applications can be achieved by adjusting the values of $\alpha$ and $\beta$ to reduce the overall cost and satisfy different overhead constraints.

- **Learning rate:** Figure S2A illustrates the effect of the learning rate on the convergence performance of the MOCI strategy. It can be seen that training agent with smaller learning rates converges more slowly, while larger learning rates lead to unstable cumulative rewards and prevent agent from exploring optimal strategies. In order to achieve a reasonable balance between stability and convergence speed, 0.0001 was experimentally chosen as the learning rate for the actor and critic networks.

- **Reuse time:** Figure S2B illustrates the effect of sample reuse time on the convergence performance of the MOCI strategy. The sample reuse time is the number of times the sample is used to train the agent. As can be seen in the figure, small sample reuse time settings lead to slow and poor convergence. As the reuse time increases, the convergence becomes faster and the strategy becomes better, but too large a sample reuse time setting instead leads to poor convergence. To strike a balance between model accuracy and training complexity, the sample reuse time is set to 20 in this experiment.

### Generalization comparison

To further validate the effectiveness of the MOCI strategy, we conduct experiments on two other popular network architectures, namely VGG11 and MobileNetV2. Their performance overheads are evaluated for a number of UEs of five.

For VGG11, four partition points were selected after the MaxPool layer. For MobileNetV2, four partition points were selected after the last batch normalization layer containing the residual block of the sampling layer, and the other settings were the same as the previous experimental settings. Figure S3 illustrates the performance comparison of the six methods for different DNN types, which verifies the scalability of the MOCI strategy for

different DNN types. Figure S3A illustrates the average inference latency under different DNN types, and it can be seen that as the complexity of DNN models increases, the processing latency of all strategies increases accordingly. However, the inference latency of the MOCI strategy is significantly lower than the other compared strategies under all DNN types, which shows its efficiency in processing complex DNN models and can better adapt to DNN models of different complexity. Figure S3B illustrates the average inference energy consumption under different DNN types, and the analysis shows that the energy consumption of all the methods increases as the complexity of the DNN model increases. However, the energy consumption of the MOCI strategy is lower than the other compared methods under all DNN types, indicating its excellent performance in energy optimization. In particular, it is still able to maintain a low level of energy consumption when dealing with complex DNN models, which is of great significance for energy consumption optimization in practical applications.

Figure S4 shows the convergence of the three strategies under the MobileNetV2 (Fig. S4A) and VGG11 (Fig. S4B) network models, from which it can be seen that the MOCI strategy converges faster than the other two algorithms and tends to be stable all the time, regardless of the network. Especially for complex network models, the MOCI strategy performs better.

We also study how different main hyperparameter settings affect the convergence performance in both DNN architectures. Comparing different learning rate and sample reuse time settings, the results are shown in Fig. S5. Figures S5A and S5B illustrate the convergence of MobileNetV2 with different learning rates and different sample reuse times, and Figs. S5C and S5D show the convergence of VGG11 with different learning rates and different sample reuse times, respectively. The results show that its parameter selection can achieve optimal results if it is consistent with the ResNet18 network. For the MobileNetV2 network, the influence of the parameters is not significant due to its lightweight characteristics, relatively low number of parameters and computational requirements, and it still maintains good performance with wider parameter settings. For the VGG11 network, its deep architecture and number of high parameters are very sensitive to the parameter settings in training, so the impact of parameter tuning on its training and performance is much more significant and requires finer tuning and control. The data show that the results of either network demonstrate the effectiveness of the MOCI strategy.

## CONCLUSION

In this article, we propose MOCI, a collaborative inference strategy for medical image diagnosis classification tasks, to address the performance and energy bottlenecks of image analysis in mobile and edge computing environments. In existing research, although some approaches try to optimize the inference efficiency of medical image analysis, they often do not fully consider the unique challenges of medical imaging tasks, such as the high-dimensional nature of data, the complexity of feature extraction, and the stringent requirements for real-time performance and accuracy. Therefore, we design a collaborative

inference strategy that can effectively balance computational resources, data transfer and energy efficiency from the characteristics of medical imaging.

Specifically, we first compress the intermediate features through coding and quantization techniques, which preserve the important information of medical images while significantly reducing the amount of intermediate data transmitted. Experimental results show that inference delay and energy consumption are significantly reduced with less than a 2% loss in classification accuracy. In addition, for the task offload scheduling problem in a multi-user environment, we further employ a MARL approach that introduces the Wasserstein distance and the LSTM network. The introduction of Wasserstein distance helps to optimize the transition between old and new strategies, improving the stability of training and avoiding the common problem of strategy oscillation in reinforcement learning. LSTM networks, on the other hand, solve the problem of temporal dependency in task scheduling, allowing offloading decisions in the middle of multiple tasks to take full account of historical information and environmental changes, effectively reducing computational and communication overheads. By combining these two parts, the MOCI strategy not only shows significant advantages on different DNN architectures, but also flexibly adapts the inference process according to different user requirements and computational resources, finding a balance between classification accuracy, latency and energy consumption.

It is worth noting that this article is a strategy study under single-edge servers, but how to further optimize resource scheduling, task offloading, and data transfer when facing more complex medical imaging tasks and extreme network environments will be the focus of future research. Future work will focus on DNN partitioning and task offload scheduling in multi-server scenarios, exploring other optimization techniques to further enhance MEC system performance. In addition, given the importance of patient privacy issues during task offloading, the research will delve deeper into privacy protection related to DNN partitioning and task offload scheduling.

### Funding

This article is supported by the Collaborative Innovation Major Project of Zhengzhou under Grant No. 20XTZX05015. The funders had no role in study design, data collection and analysis, decision to publish, or preparation of the manuscript.

### Grant Disclosures

The following grant information was disclosed by the authors:
Collaborative Innovation Major Project of Zhengzhou: 20XTZX05015.

### Competing Interests

The authors declare that they have no competing interests.

## Author Contributions

- Shiqian Zhang conceived and designed the experiments, performed the experiments, analyzed the data, performed the computation work, prepared figures and/or tables, authored or reviewed drafts of the article, and approved the final draft.
- Yong Cui conceived and designed the experiments, performed the computation work, authored or reviewed drafts of the article, and approved the final draft.
- Dandan Xu analyzed the data, prepared figures and/or tables, and approved the final draft.
- Yusong Lin performed the experiments, authored or reviewed drafts of the article, and approved the final draft.

## Data Availability

The data is available at Zenodo: zsq-github. (2024). zsq-github/MOCI: MOCI (v1.0.0). Zenodo. https://doi.org/10.5281/zenodo.13738800.

The Chest CT-Scan images Dataset is available at Tianchi and Zenodo:

- https://tianchi.aliyun.com/dataset/93929.
- SunneYi. (2025). Chest CT-Scan images Dataset [Data set]. Zenodo. https://doi.org/10.5281/zenodo.14759927.

## Supplemental Information

Supplemental information for this article can be found online at http://dx.doi.org/10.7717/peerj-cs.2708#supplemental-information.

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
