# Peer review of "A collaborative inference strategy for medical image diagnosis in mobile edge computing environment"

_PeerJ Computer Science, doi:10.7717/peerj-cs.2708_

## Round 0.1 · original submission · Major Revisions

Your work has merits, but also has some issues. Please carefully consider the comments of the reviewers and revise it accordingly.

Reviewer 1 ·

Basic reporting

The introduction lacks clarity and sufficient informative explanation of the problem statement.

2. The literature review does not provide a critical comparison of related work. Provide a more detailed analysis of the strengths and weaknesses of previous methods is necessary.

Experimental design

3. Methodology is unclear in describing how the proposed model optimizes the balance between energy consumption and accuracy. The equations need clear explanations, particularly the quantization techniques. The LSTM structure is introduced for stability, yet no specific performance metrics are provided regarding how much improvement in latency and energy consumption the LSTM offers.

4. There is no information about the hyperparameter tuning in the experiment section. More details are needed on how parameters like compression rate were selected and optimized.

5. Experimental setup does not explain why only three DNN models (ResNet18, VGG11, MobileNetV2) were chosen. Provide more clear discussion on the results and include a quantitative comparison with similar methods from literature studies.

6. It is mentioned that feature compression as a key contribution, but how does the proposed multi-layer autoencoder method compare in computational complexity to other state-of-the-art compression methods in terms of both memory usage and latency?

7. While you highlight the effectiveness of Wasserstein distance to constrain old and new strategies, provide experimental data showing how much this technique reduces the divergence between strategies.

8. The table of results in the experimental section is confusing, and it’s unclear how the compression rate was determined for each DNN model. Clarify the criteria used to set these rates.

Validity of the findings

9. In the results section, figures are hard to interpret and unclear. The legends and axis labels should be more descriptive, and the use of colour should be reconsidered for clarity.

10. The conclusion is too brief and does not adequately summarize the practical implications of the study's results. It should also revisit the research questions.

Reviewer 2 ·

Basic reporting

1. Abstract
Comments:
• The abstract is clear but would benefit from a stronger problem statement to emphasize the research gap and need for this approach.
• The explanation of the proposed strategy (MOCI) is well-summarized, but details like the significance of "Wasserstein distance" or "LSTM network" may not be immediately clear to all readers.
• Conclude the abstract with a statement on the potential implications or applications of MOCI in real-world MEC scenarios.
2. Introduction
• Consider restructuring this section to introduce the specific limitations of existing DNN models, MEC solutions, and then directly state the primary contribution of the paper.
• Emphasize the novelty of MOCI by comparing it with the most recent advancements in collaborative inference models.
3. Related Work
• However, the section could benefit from a clearer differentiation between MOCI and the methods discussed, as it would highlight the novelty and relevance of the study.
• Add a comparison table summarizing the key aspects of each related method and highlighting MOCI’s distinctive features.
• Briefly comment on the limitations of each approach discussed, which MOCI seeks to overcome.
4. System Overview and Problem Modeling
• Incorporate a flowchart or system diagram to illustrate the task distribution, model partitioning, and data transmission between UEs and edge servers.
• Clarify assumptions and parameter selections within the problem formulation for clearer understanding.
5. System Design
• Explain specialized terms like “multi-intelligent body” or provide a brief background on relevant algorithms (e.g., PPO) for non-expert readers.
• Consider a flow diagram to visualize the steps in the MOCI strategy workflow.
6. Experiments
• Briefly explain why ResNet18, VGG11, and MobileNetV2 were chosen as experimental models.
• Provide a rationale for the choice of parameter values, such as CPU settings or dataset selection, and explain how they reflect real-world MEC environments.
7. Results and Discussion
• Discussion on parameter sensitivity and generalization to different DNN architectures is strong, but insights could be expanded.
• Include a discussion on the potential trade-offs between latency and energy savings, particularly for different MEC applications.
• Provide a comparative analysis with alternative strategies, especially in terms of computational and energy efficiency.
8. Conclusion
• Add potential applications of MOCI and areas for future research, such as adapting MOCI for more complex imaging tasks.
• Summarize any observed limitations of MOCI in terms of real-world scalability or technical constraints that were beyond the study's scope.
General Comments
• The technical language is mostly clear, but some sentences are complex and could be simplified for better readability.
• The technical descriptions, especially for reinforcement learning and DNN segmentation, are in-depth. However, simplifying and clarifying certain technical terms would make the paper more accessible to a broader audience.

Experimental design

No Comment

Validity of the findings

No Comment

Additional comments

No Comment

---

## Round 0.2 · accepted · Accept

Thanks for your efforts to improve the work. The previous reviewers have not responded to the invitation to re-review. I have evaluated your revision myself, and I believe the current version has issued the concerns of the reviewers.